# A Sensitive Capacitive Biosensor for Protein a Detection Using Human IgG Immobilized on an Electrode Using Layer-by-Layer Applied Gold Nanoparticles

**DOI:** 10.3390/s22010099

**Published:** 2021-12-24

**Authors:** Kosin Teeparuksapun, Martin Hedström, Bo Mattiasson

**Affiliations:** 1Science Program, Department of General Education, Faculty of Liberal Arts, Rajamangala University of Technology Srivijaya, Songkhla 90000, Thailand; kosin.t@rmutsv.ac.th; 2Division of Biotechnology, Lund University, P.O. Box 124, 221 00 Lund, Sweden; martin.hedstrom@biotek.lu.se

**Keywords:** capacitive biosensor, layer-by-layer, gold nanoparticle, thiourea, protein A

## Abstract

A capacitive biosensor for the detection of protein A was developed. Gold electrodes were fabricated by thermal evaporation and patterned by photoresist photolithography. A layer-by-layer (LbL) assembly of thiourea (TU) and HAuCl_4_ and chemical reduction was utilized to prepare a probe with a different number of layers of TU and gold nanoparticles (AuNPs). The LbL-modified electrodes were used for the immobilization of human IgG. The binding interaction between human IgG and protein A was detected as a decrease in capacitance signal, and that change was used to investigate the correlation between the height of the LbL probe and the sensitivity of the capacitive measurement. The results showed that the initial increase in length of the LbL probe can enhance the amount of immobilized human IgG, leading to a more sensitive assay. However, with thicker LbL layers, a reduction of the sensitivity of the measurement was registered. The performance of the developed system under optimum set-up showed a linearity in response from 1 × 10^−16^ to 1 × 10^−13^ M, with the limit detection of 9.1 × 10^−17^ M, which could be interesting for the detection of trace amounts of protein A from affinity isolation of therapeutic monoclonal antibodies.

## 1. Introduction

The recent expansion of biopharmaceutical processes has created a large demand for rapid and sensitive analytical techniques for the detection of trace concentrations of impurities [1]. Previously, it was sufficient to state the percentage of the pure product; however, recent concerns have arisen about the identity and quantification of the remaining impurities. Host-cell proteins [2,3], endotoxins [4], nucleic acids, and viruses [5] constitute such impurities, together with reagents from the downstream processing. Contamination of monoclonal antibody preparations by protein A (PA) leaching from the affinity adsorbent is such an example [6,7]. Products contaminated with PA can cause immunogenic [8] and mitogenic effects [9,10]. Hence, the high purity of immunoglobulin for clinical applications is required [6].

Quantification of PA, which might be present at very low concentrations, is usually a tedious and cumbersome process [11]. To be able to detect the trace amounts of PA, a pre-enrichment step, for example, the use of another affinity step aimed at capturing the leaking ligands, might be required before the analysis [12]. Although the affinity sensor with immobilized IgG is readily available for the detection of leaking PA from the affinity column used for the isolation of monoclonal antibodies, an assay set-up might nevertheless remain complicated, even if the analytical sensitivity is high enough to detect PA at low concentrations [1]. Enzyme-Linked Immunosorbent Assay (ELISA) is a method often used for the analysis of protein A [11,12]. Unfortunately, this method requires labeling of primary antibodies and highly trained personal, and it is time consuming and sub-optimal for measuring at low concentrations.

Capacitive biosensors may overcome these problems. This technique is highly sensitive [1,13,14,15]. The assaying principle is simple and does not require the use of labeled reagents. The development and applications of capacitive biosensors have been widely reported [1,16,17,18,19]. As in other kinds of immunosensors, the analytical sensitivity of the sensor depends on the binding of target molecules to the antibodies immobilized on the transducer surface. As a consequence, immobilization strategies of the antibody are of great importance, and the sensitivity is believed to be dependent on the orientation and amount of antibody immobilized on an electrode surface [13,14,20,21,22,23].

Self-assembled monolayers (SAMs) of alkanethiols are often employed for the immobilization of antibodies in capacitive biosensors [15,16,17,18,24,25]. However, the preparation is time consuming, as the complete formation of a monolayer requires several hours. Application of gold nanoparticles (AuNPs) in biosensor development is of technological importance, since they have played an increasingly important role for improving the performance of the sensors [13,14,26,27,28]. The AuNPs can be applied directly for increasing the available surface area for immobilization, which could enhance the immobilization yield of biomolecules such as antibodies [13,14,23], or it can be pre-functionalized to enhance the immobilization of biomolecules [28,29]. With the use of AuNPs, one of the advantages is that biomolecules still remain active when they adsorb on AuNPs [30,31].

The application of AuNPs for immobilization of biomolecules can be done in several ways. The simplest strategy is the direct coupling of AuNPs to the transducer surface on which functional groups such as -NH_2_ [13,14,32] or -SH [33,34,35] were first introduced. The AuNPs can bind to such functional groups via a chemisorption process [13]. Several studies have reported that the use of AuNPs can enhance the immobilization of antibodies and thus increase the sensitivity of the sensor [13,14,36]. However, finding a suitable application of AuNPs for the immobilization method is still a challenge, especially for the development of highly sensitive assays. An alternative is to create a preparation of immobilized biomolecules via a layer-by-layer (LbL) assembly technique. This approach has attracted attention and has been demonstrated to offer suitable model surfaces for antibody immobilization [37,38,39]. The technique of applying nanoparticles using the LbL concept opens great opportunities to obtain a surface with several advantages, such as a wide choice of materials, simplicity, and precise control of thickness and layer composition [40,41,42,43], and it has been proven to be a promising technique for the immobilization of biomolecules [33,41,42,44,45].

In this paper, a layer-by-layer (LbL) assembly of TU and AuNPs was prepared using self-assembled monolayer (SAM) techniques and a chemical reduction method. The LbL layers were used for the immobilization of human IgG as a specific probe for the detection of protein A in a flow injection capacitive biosensor system.

The number of LbL layers (LbL_n_) and their sensitivities for protein A detection were investigated and compared to those obtained from a conventional self-assembled monolayer (SAM) of TU with glutaraldehyde cross-linking. The effect on analytical performance of capacitive immunosensors as a function of the number of layers was investigated and discussed.

## 2. Materials and Methods

### 2.1. Material

Silicon wafers (Ø = 3″, resistivity 1.0–10 Ω/cm^2^, thickness 320–370 µm) from Addison Engineering (San Jose, CA, USA) were used as the substrate for fabrication of gold electrodes. Chloroauric acid (HAuCl_4_) and thiourea (TU; NH_2_CSNH_2_) were purchased from Sigma-Aldrich^®^ (Steinheim, Germany). Human gamma immunoglobulin (human-IgG) was purchased from Octapharma AB (Stockholm, Sweden). Protein A was a kind gift from Novozymes Biopharma (Lund, Sweden). All other chemicals were of analytical grade. Buffers were prepared with water treated with a de-ionizing system and processed by a reverse osmosis step with a Milli-Q system from Millipore (Bedford, MA, USA). This water is called Milli-Q water in the following text. Prior to use, all buffers were filtered through a Millipore filter (pore size 0.22 µm) and degassed for 1 h.

### 2.2. Methods

#### 2.2.1. Fabrication of Gold Electrode

Gold electrodes were fabricated by thin film deposition using thermal evaporation and patterned by photoresist lithography and a wet etching. The electrodes were fabricated by coating thin layers of first 50 nm of chromium and then 200 nm of gold on a silicon wafer surface using a thermal evaporator E306 (Edwards, Sussex, UK). The Cr layer was applied to enhance the adhesion of Au. The pattern of electrodes was made via photoresist lithography [46]. First, the Cr-Au coated on the silicon wafer surface was coated with positive photoresist by holding a wafer on a spinner chuck under vacuum; the photoresist was coated to uniform thickness by spin coating at 3900 rpm for 60 s. Solvents from the photoresist coating were removed by soft baking at 80 °C for 20 min. Thereafter, a photo mask was aligned with the Cr-Au-photoresist coated wafer, and the photoresist was exposed through the pattern on the mask with a high intensity of ultraviolet (UV) light. The chemical structure of the photoresist exposed to the UV light was changed and could be removed by the developer (MicropositTM 351 developer 1: 4 Milli-Q water). The Au and Cr layers were then etched by gold and Cr etching solution, respectively. This step resulted in a pattern of Au electrode on the silicon surface; the silicon plate was then cut to a dimension of 7 × 23 mm with a dicing saw. In this case, the photoresist remained coated on the surface to protect the Au layer from the dust from the cutting step. Finally, the photoresist was removed by soaking in acetone and subsequently ethanol. It was then dried with pressurized air and kept in a desiccator at 4 °C.

Surface properties of the fabricated electrodes were characterized using Atomic Force Microscopy (AFM) from NTEGRA (Zelenograd, Russia). The image was taken in tapping mode using a silicon cantilever of the type NT-MDT (Molecular Device and Tools for Technology (NT-MDT)) from NTEGRA.

#### 2.2.2. Preparation of Layer-by-Layer (LbL) of TU and AuNPs and Human IgG Immobilization

Layer-by-layer (LbL) of self-assembled TU and AuNPs was created on the surface of the Au electrode and used as a platform for the immobilization of human IgG. The electrode surface was first cleaned with freshly prepared Piranha solution (3:1 mixture of concentrated H_2_SO_4_ and 30% *w*/*w* H_2_O_2_) through sonication for 15 min in order to remove dust or any possible residuals from the fabrication process. The electrodes were then rinsed several times with Milli-Q water, dried with nitrogen gas, and placed in a plasma chamber for 20 min (Model PDC-3XG, Harrich, NY, USA) to remove possible remaining impurities and contaminants from surfaces. The preparation of LbL of AuNPs on the assembly of TU was modified from Tang et al., 2008 [33]; this is described in Figure 1a–e. For one layer (LbL_1_) (Figure 1a), the preparation consisted of three steps. First, cleaned Au electrodes were immersed in 250 mM TU solution at room temperature for 2 h (Figure 1a(1)). In this step, self-assembled TU monolayer had formed on the Au surface (Au-TU). After rinsing with Milli-Q water, electrodes were immersed in 100 mM HAuCl_4_ solution in the dark for 2 h with gentle mixing (Au-TU_1_-Au_1_). During this step, the [AuCl_4_]^−^ ions formed a complex to the NH_2_ groups of self-assembled TU on the electrode (Figure 1a(2)). Afterwards, the electrode with [AuCl_4_]^−^-NH_2_ complexes was immersed in 100 mM NaBH_4_ containing 300 mM NaOH under gentle mixing of the solution (Figure 1a(3)). This step led to the reduction of Au^3+^ to neutral AuNPs, and the change of color from yellow to light red was observed at the electrode surfaces.

To obtain more layers (Figure 1b–e), an electrode with [AuCl_4_]^−^-NH_2_ complex (Au-TU_1_-A_U1_) was then again immersed in TU (step 1) after exposure to HAuCl_4_ (step 2) solution. The process was repeated for the assembly of TU and [AuCl_4_]^−^ ions to obtain the desired number of layers. At the end, the electrodes were immersed in NaBH_4_, similar to the case of one layer (LbL_1_), and these electrodes were named as LbL_n_, where n is the number of layers.

The TU-AuNPs-LbL_n_ modified electrodes were then rinsed intensively with 100 mM sodium phosphate buffer, pH 7.2, and dried with pure nitrogen gas. Then, 20 µL of 1.65 mg/mL human IgG was dropped on the electrode surface and left overnight at 4 °C. The antibodies bound randomly to the AuNPs surface through chemisorption. The electrode with immobilized human IgG was kept at 4 °C until further use. The number of layer-by-layer assemblies were varied, and their analytical characteristics for the detection of protein A were evaluated.

Immobilization of human IgG was also studied with conventional self-assembled thiourea [47] in order to evaluate the performance of the LbL assembly for the enhancement of antibody immobilization and the resulting sensitivity. Briefly, the cleaned gold electrodes were immersed in 250 mM TU solution for 2 and 24 h. The TU modified electrodes were then treated with (5% *v*/*v*) glutaraldehyde in 10 mM sodium phosphate buffer at pH 7.2 for 20 min to introduce aldehyde groups on the electrode surface. Human IgG (1.65 mg/mL) was then immobilized on the activated surface. Prior to use, all aldehyde groups that did not couple with human IgG were treated with 0.1 M ethanolamine at pH 8.00 for 30 min. Before the analysis, prior to placement into the detection flow cell of the capacitive measuring system, the electrodes were immersed in 10 mM 1-dode canethiol ethanolic solution for 20 min to cover bare parts of the Au surface, resulting in a completely insulated surface. The sensitivities of electrodes prepared by LbL and conventional SAM techniques were evaluated, and the electrode that gave the best performance was used for further study of analytical performances, linearity, and detection limits.

#### 2.2.3. Characterization of LbL Modified Electrodes by Cyclic Voltammetry

The cyclic voltammograms of electrodes after modification with layer-by-layer TU and AuNPs can give useful information. This experiment was conducted to prove if the AuNPs were successfully reduced at the last step of the preparation. In theory, the electrode modified with AuNPs will enhance the electron transfer, and this will result in higher redox peaks of cyclic voltammogram of the redox couple, e.g., potassium ferricyanide. Thus, the more layers, the higher the redox peaks. The cyclic voltammetry measurement was performed using a potentiostat/galvanostat/Autolab (PGSTAT 12, The Netherlands) equipped with GPES 4.8 software. Electrochemical measurements involved three electrodes: an Au electrode, either unmodified or covered by different layers, as working electrode; and a custom-made silver-silver chloride (Ag/AgCl) and a platinum wire, serving as reference and auxiliary electrode, respectively. All cyclic voltammograms were recorded in 50 mM potassium ferricyanide K_3_[Fe(CN_6_)] containing 0.1 M KCl by sweeping the potential range between −300 and +800 mV at a sweep rate of 100 mV/S.

#### 2.2.4. Quantification of Immobilized Human IgG

The amounts of human IgG immobilized on electrodes prepared with different numbers of layer-by-layer (LbL_1_-LbL_4_) and typical SAMs were calculated after the human IgG was immobilized on the electrodes. The electrode surfaces were washed with 10 mM sodium phosphate buffer at pH 7.2. The washing volume was collected and used for the quantification of human IgG. The immobilized amount was calculated by subtracting the amount of human IgG found in the washing fraction from the initial loading amount. The concentrations of human IgG in washing fractions were quantified by Bicinchoninic acid (BCA) Protein Assay. Ten microliters of BSA protein standards (0.20, 0.40, 0.60, 0.80, and 1.0 mg/mL) and samples were added with 200 µL of BCA working reagent (Bicinchoninic Acid Protein Assay kit, BCA-1, Sigma) (1:50 ratio of CuSO_4_ to BCA) in a 96-well plate. After incubation at 37 °C for 30 min, the absorbance at 562 nm of each solution was recorded. Human IgG concentration in each sample was determined by comparing the absorbance of samples to the standard curve and multiplying with the dilution factors.

#### 2.2.5. Capacitance Measurement

An electrode with immobilized human IgG was inserted as the working electrode in a specially constructed flow cell equipped with four electrodes and a dead volume of 10 µL [10]. A platinum foil and platinum wire served as auxiliary and reference electrodes, respectively. An external reference electrode (Ag/AgCl) was placed in the outlet stream. A fast data acquisition unit (575 measurement and control system, Keitley Instrument, Cleveland, OH, USA) was connected between the potentiostat and a personal computer. The carrier buffer was pumped by a peristaltic pump (M-312 Gilson, France) at a controlled flow rate of 100 µL/min. Protein A was injected into the flow stream via a 250 µL sample loop, and the sensor was regenerated with pre-optimized regeneration buffer (25 mM glycine-HCl, pH 2.4). The capacitance measurement, which is based on potentiostatic step, was performed in the same way as described in previous reports [11,12,13,14].

#### 2.2.6. Regeneration of the Electrode

In order to obtain a reproducible capacitive measurement, a complete regeneration of the surface between the injections of PA must be obtained. The efficiency of regeneration was studied by repeated injections of a standard PA at the same concentration (1.0 × 10^−14^ M) followed, after reading the capacitance change, by a pulse of 25 mM glycine-HCl at pH 2.4 as regeneration solution. The study was done over 2 days (20 injections of PA and regeneration solution/day). To test the performance of the regeneration solution, the percentage residual activity was calculated (Equation (1)) from the capacitance change (∆C) as a consequence of the binding between PA and human IgG before (∆C_1_) and after regeneration (∆C_2_) [9,10,42].
(1)$$\%\;\mathrm{residual}\;\mathrm{activity}=\left(\frac{\Delta C_{2}\times 100}{\Delta C_{1}}\right)$$

#### 2.2.7. Reproducibility of Electrode Preparation

The reproducibility of the electrode preparation was studied using the optimum condition obtained from Section 2.2.2. Four electrodes were prepared, and their performances were investigated and compared using capacitive measurement.

## 3. Results

### 3.1. AFM Images

Atomic Force Microscopy was used to image the surface of the electrode. The two-dimensional AFM image of the Au electrode fabricated by the thermal evaporation technique and patterned with photoresist lithography is shown in Figure 2a. The image exhibits a uniform pattern, corresponding to a spherical gold cluster, with a diameter of approximately 50 nanometers (nm). The three-dimensional AFM image of the Au electrode is shown in Figure 2b. This image was used for the calculation of surface roughness of the electrode, and it was found to be 2.8 nm.

### 3.2. Cyclic Voltammetry

Theoretically, the attachment of AuNPs on the electrode surface can enhance the ability of the electron transfer due to the large surface area of AuNPs. In this work, the electrodes were modified with AuNPs via an LbL assembly of TU and AuNPs. It is expected that the electrode modified with more layers of LBL will have more AuNPs and thus better electron transfer. This hypothesis was tested by cyclic voltammetry in ferricyanide solution. For a bare Au electrode, when the potential was swept from −300 to +800 mV, the anodic peak due to the oxidation of [Fe^2+^(CN_6_)]^4−^ to [Fe^3+^(CN_6_)]^3−^ was observed. Once the potential was swept back, the cathodic peak of the reduction of [Fe^3+^(CN_6_)]^3−^ to [Fe^2+^(CN_6_)]^4−^ was observed. The redox peak of the bare Au electrode is shown in Figure 3a.

For the electrode modified with TU and AuNPs, after treating with NaBH_4_, the electrodes modified with LbL_1–4_ were recorded for their cyclic voltammograms. As expected, all cyclic voltammograms of LbL-modified electrodes gave higher redox peak currents as compared to what was observed for the bare Au (Figure 3b–e). This may be due to the fact that AuNPs attached on electrode surface can enhance the electron transfer mechanism between the redox species and electrode surface. The redox peak current is slightly increased when the LbL is increased from one (LbL_1_) to two (LbL_2_) layers. The increase is further seen for three layers (LbL_3_) and just slightly further increase in the case of (LbL_4_). For these electrodes, after the human IgG was immobilized, the redox peaks were again decreased. Finally, the surfaces of electrodes were completely insulated after being immersed in 10 mM 1-dodecanethiol (data not shown).

The cyclic voltammogram for the immobilization of human IgG using SAM TU and glutaraldehyde treatment was also investigated. As expected, the redox peak of the bare Au electrode was decreased after the SAM formation of TU formed on the electrode surface. The electrode treated with TU for 24 h showed a higher degree of insulation as compared to the one that had been treated for only 2 h, indicating a more complete formation of SAM, with longer assembly time. However, the redox peaks of these electrodes were decreased when treated with glutaraldehyde and further decreased after human IgG was coupled. As in LbL-modified electrodes, the electrodes prepared with SAM of TU were completely insulated when treated with 1-dodecanethiol. The electrochemical results demonstrated that the modified electrode was highly insulated and could be further employed for the detection of protein A in a capacitive flow injection system.

### 3.3. Amount of Immobilized Human IgG

The amount of immobilized human IgG was determined for the electrodes modified with different numbers of layer-by-layer of TU and AuNPs and those prepared by conventional SAM of TU, with glutaraldehyde as cross-linked agent. Among the electrodes prepared by LbL techniques, the amount of human IgG immobilized on electrode surfaces prepared by a different number of layers was 0.84 ± 0.02, 0.90 ± 0.02, 0.95 ± 0.01, and 0.99 ± 0.01 mg/mL for LbL_1_, LbL_2_, LbL_3_, and LbL_4_ modified electrodes, respectively. These results support the fact that the more layers, the more gold nanoparticles attached to the electrode surface, which will increase the immobilization yield due to the increased surface area of AuNPs provided for the chemisorption of human IgG. The immobilized amount of human IgG for electrodes prepared with SAM was 0.73 ± 0.01 mg/mL (2 h SAM) and 0.80 ± 0.06 mg/mL (24 h SAM), which is lower than for the electrodes modified with the LbL assembly.

### 3.4. Capacitive Measurement with Human IgG Modified Electrode

Capacitive measurement is based on the theory of electrical double layer. The human IgG modified electrode resembles a capacitor when placed in an electrolyte solution. The capacitance of the modified electrode is composed of a series of capacitances, as schematically presented in Figure 4a for an electrode modified with LbL_1_.

A total capacitance (C_TOT_) has three components: (1) the capacitance of the insulating layer (C_INS_), (2) the capacitance of the human IgG layer (C_IgG_), and (3) the capacitance of the Gouy-Chapman diffuse layer (C_GC_). In this scenario, the lowest capacitance dominates the value of total capacitance; therefore, C_INS_ and C_GC_ should be made as large as possible so that the change of C_IgG_ caused by the binding of protein A can be detectable.

Capacitance is evaluated by perturbing the system with a potential step. A potentiostatic step with an amplitude of 50 mV was applied every minute, recording the current transients evoked by the potential step at a frequency of 50 kHz according to Equation (2).
(2)$$i\left(t\right)=\frac{u}{R_{s}}\exp\left[\frac{-t}{R_{s}C_{TOT}}\right]$$
where *i*(*t*) is the current in the circuit as a function of time, *u* is the applied potential, *R_s_* is the dynamic resistance of the recognition layer of human IgG, *t* is the time elapsed after the potential step was applied, and *C_TOT_* is the total capacitance measured at the electrode/solution interface. Taking the logarithm of Equation (1) initially results in an almost linear curve between the logarithm of current (*ln i*) and *t*. Using linear least-square fitting, the *C_TOT_* and *R_s_* can be obtained from the slope and intercept of the linear curve [13,14,15].

The time course diagram showing the decrease in capacitance after injection of the PA standard is depicted in Figure 4b. Prior to the analysis, the regeneration solution was injected into the system to remove any human IgG that bound weakly to AuNPs. After the stable baseline was achieved, an average value of the last five capacitance reading (5 min), just before the loading of protein A, was calculated and considered as original baseline (C_1_). After 2–3 min of PA injection, the PA started to bind to the Fc part of the human IgG, and this resulted in the decrease of capacitance. After the capacitance was registered for 15 min, the last five readings were averaged and considered as a baseline after binding occurred (C_2_). The change in capacitance ∆C was then calculated as C_1_-C_2_ and registered for each concentration of PA. The ∆C was then calculated per surface area of working electrode and reported as −nF/cm^2^. Regeneration solution was injected after each assay to break the interaction between human IgG and protein A, resulting in a reusable surface.

### 3.5. Sensitivity of the Biosensor Electrodes

The sensitivity of the biosensor electrodes was investigated by the injection of standard solutions of PA at concentrations ranging from 1.0 × 10^−17^ to 1.0 × 10^−11^ M. The capacitance change (−nF/cm^2^) obtained from each electrode was plotted against the logarithm of concentration (log M) to evaluate the linearity. The sensitivity (−nF/cm^2^/log M), the slope of the linear range, was then calculated as shown in Table 1.

As can be seen in Table 1, the electrode modified with SAM of TU for 2 h and with antibodies immobilized via cross-linking using glutaraldehyde has lowest sensitivity (2.31 ± 0.23 −nF/cm^2^/log M). This result is correlated to the amount of the immobilized antibodies, which is the lowest among the studied electrodes. For an electrode prepared with SAM for 24 h, the sensitivity increased to 5.11 ± 0.23 −nF/cm^2^/ log M. This sensitivity is equal to that of the electrode prepared with LbL_1_ (5.48 ± 0.31 −nF cm^2^/log M). However, the preparation time of LbL_1_ is only 5 h, which is much shorter than in the case of SAM, which required at least 24 h. When the number of layers was increased to two layers (LbL_2_), the sensitivity also increased to 7.71 ± 0.30 −nF/cm^2^/log M. However, the sensitivity did not increase when the number of layers was increased to three layers (LbL_3_): the sensitivity of LbL_3_ was 7.66 ± 0.43 −nF/cm^2^/log M. No significant change was observed when the number of layers was increased to four layers (LbL_4_) (7.69 ± 0.37 −nF/cm^2^/log M). When the layers were increased to five (LbL_5_), the sensitivity decreased to 7.69 ± 0.37 −nF/cm^2^/log M.

### 3.6. Analytical Characteristics

#### 3.6.1. Linear Range and Limit of Detection

The electrode prepared by LbL_2_ was chosen as optimum and used to evaluate the analytical performance of the sensor. A linear relationship between 1.0 × 10^−16^ and 1.0 × 10^−13^ M was observed (Figure 5), with a limit of detection of 9.1 × 10^−17^ M based on IUPAC Recommendation 1994 [48].

#### 3.6.2. Regeneration of Electrodes

It is desirable to regenerate the surface of the sensor and be able to use the sensor for several cycles of analysis. The regeneration is thus a critical step, as it might affect the activity of the biological sensing element and thus result in loss of sensor sensitivity. In this work, the regeneration was evaluated after dissociating the binding of PA to human IgG using glycine-HCl buffer (25 mM, pH 2.4). The capacitance change (−nF/cm^2^) from the injection of the same concentration of standard PA was investigated over a period of 2 days (20 injections per day). The regeneration procedure allowed the dissociation of the PA-human IgG interaction, evidenced by the returning of the capacitance baseline. The capacitance change was registered and calculated as percentage residual activity by comparing to the first injection. The electrode can be reused up to 30 times, with an average residual activity of 96 ± 2% according to Figure 5-insert. The storage stability of electrodes was also investigated. Electrodes were kept at 4 °C in a closed Petri dish filled with nitrogen gas, and the sensitivities were tested after 14 days. The results showed that the sensitivity of the electrode decreased by only 1.52 ± 0.90%. Thus, the prepared electrode provided good storage stability.

#### 3.6.3. Reproducibility of Electrode Preparation

The reproducibility between four different electrodes prepared with LbL_2_ was investigated. The sensitivities (slope) of the calibration curves of four electrodes were investigated and found to be 7.71 ± 0.47, 7.69 ± 0.50, 7.62 ± 0.31, and 7.79 ± 0.34 −nF/cm^2^/log M. These results show that the preparation of electrodes using layer-by-layer of TU and AuNPs provides an immobilization strategy with good reproducibility.

## 4. Discussion

The production of monoclonal antibodies is important for biopharmaceutical industries. Protein A affinity chromatography is widely used for commercial purification of monoclonal antibodies due to high selectivity, capacity, and product purity. Leaching of this ligand from the affinity resin into the elution pool of a monoclonal antibody is a major problem in biopharmaceutical production, especially for therapeutic antibodies, since PA can cause health problems. Thus, it is a regulatory expectation that leached PA should be cleared during the purification of antibodies for human use. PA impurities might present in trace amounts. Thus, it is essential to have a method that can accurately quantify the amount of PA leakage.

In this study, we developed a capacitive biosensor for the detection of PA. The experiment focused on the enhancement of sensitivity of the sensor by increasing the amount of immobilized biomolecules (human IgG). The immobilization of human IgG was carried out using a layer-by-layer assembly of thiourea (TU) and chloroauric acid (HAuCl_4_), followed by chemical reduction with sodium borohydride (NaBH_4_). The human IgG was randomly adsorbed on the AuNPs via chemisorption interaction, providing a probe with some Fc parts of human IgG available for protein A detection. The lengths of the layer-by-layer probe were varied by a controlled number of cycles for TU and HAuCl_4_ assemblies. As shown in the results, the increase in the number of layers (LbL_1_, LbL_2_) can enhance the concentration of immobilized human IgG and thus enhance the sensitivity for PA detection. This phenomenon can be explained by the increased PA binding sites, with increased concentration of immobilized human IgG with LbL of TU and AuNPs. The dramatic increase in sensitivity may be explained by nano-environmental effects at the sensor surface. It is reported that the binding efficiency between antibodies (Abs) and antigens (Ags) is influenced by the nanoenvironment around them. The concentration of Abs on a sensor surface achieves a high concentration of these biomolecules in close vicinity to the sensor surface. Theoretically, the concentration of the Ab localized at the electrode surface could be calculated. The assumption that Ab is a cube with 40 Å of each side would give 0.6 × 10^14^ molecules per cm^2^. Therefore, an Ab layer with a thickness of 40 Å would translate into a volume per cm^2^ of 4 × 10^−7^ cm^3^, ultimately giving a local concentration of approximately 25 mM. This high local concentration of antibodies provides firm binding [49].

It is clearly seen in this work that the ultrasensivity of the capacitive sensor was achieved by using AuNPs. The nanoparticles considerably improve sensitivity by increasing the amount of human IgG immobilized on the electrode surface. Table 2 shows the comparison of the detection range and limit of detection (LOD) of the present study with related works focused on layer-by-layer immobilization techniques.

Although the use of AuNPs added layer-by-layer contributes to the higher sensitivity of capacitive sensor, the lengths of the probe play a key role in capacitance signal. Since capacitive assays are based on registering the capacitance between the biosensor electrode and the layer of counter ions that will become displaced when Ab-Ag interaction takes place, this means that a decrease in capacitance depends on the amount of bound material that will displace the counter ions.

The experimental results reveal that at a certain distance from the electrode surface, the sensitivity of the capacitive sensor starts to decrease, and it is not possible to detect any further decrease in signal if one has passed out of the layer that is suitable for the measurements. This phenomenon was observed in this work when human IgG was immobilized with four and five layers of TU-AuNPs probes. The explanation of the observed characteristics might be a limitation of the capacitive sensor when the distance of the diffuse layer is too far from the sensor surface. The theoretical background of the capacitive measurement is based on the theory of electrical double layer (EDL). The electrode with immobilized antibodies behaves like a capacitor composed of two metal sheets separated by a dielectric material when it is brought into contact with the electrolyte solution. Theoretically, the modified electrode used in the capacitive biosensor needs to be thin and insulated. The structure of EDL depends on the interactions between the charged electrode and solvated ions, which are governed by electrostatic force, and is independent from chemical properties. Therefore, when modifying electrodes, if the modified and/or insulating layer is too thick, this will result in a weak electrostatic interaction and formation of a poor electrical double layer. This behavior might influence the capacitive measurement, as the electrode does not behave as a capacitor, and therefore the use of the RC circuit for measuring the capacitance might not be valid. Therefore, the sensitivity of the measurement might be decreased, as can be seen in the studies of LbL-modified electrodes presented in this work (LbL_4_ and LbL_5_). Therefore, it is important in capacitive biosensor construction that the electrode be insulated and the modified layer be designed as thin as possible.

The ultrasensitivity of the capacitive sensor obtained in this work made it possible to perform analyses of trace amounts of PA. Furthermore, since the sensitivity is high, it is possible to dilute the sample several orders of magnitude before analysis, thus eliminating the matrix interferences.

## 5. Conclusions

In conclusion, the present work demonstrates the development of capacitive biosensors aiming for a detection of protein A. The study was focused on a layer-by-layer assembly of TU and HAuCl_4_, with subsequent immobilization of human IgG. The results show that the use of the LbL strategy can enhance the immobilization efficiency of human IgG and thus increase sensitivity of the measurement. There are arguments surrounding the extreme sensitivity of the capacitive biosensor, and the results in this experiment support the fact that one parameter that provides ultra-sensitivity is the amount of the immobilized ligand (e.g., human IgG in this case). However, there are also limitations of the capacitive measurement; the measurement of capacitance over the electrode surface is effective only for a certain distance between the electrode and bulk solution. If the biosensor electrode were constructed in a way that the diffuse layer is too far from the sensor surface, the sensitivity of the measurement would be reduced. The analytical performance of the electrode under optimum conditions (LbL_2_) provides three orders of magnitude from 1.0 × 10^−16^ to 1.0 × 10^−13^ M, with a highly sensitive limit of detection (9.1 × 10^−17^ M). Moreover, the electrode can be reused up to 30 times with good reproducibly of the electrode preparation.

## Figures and Tables

**Figure 1 sensors-22-00099-f001:**
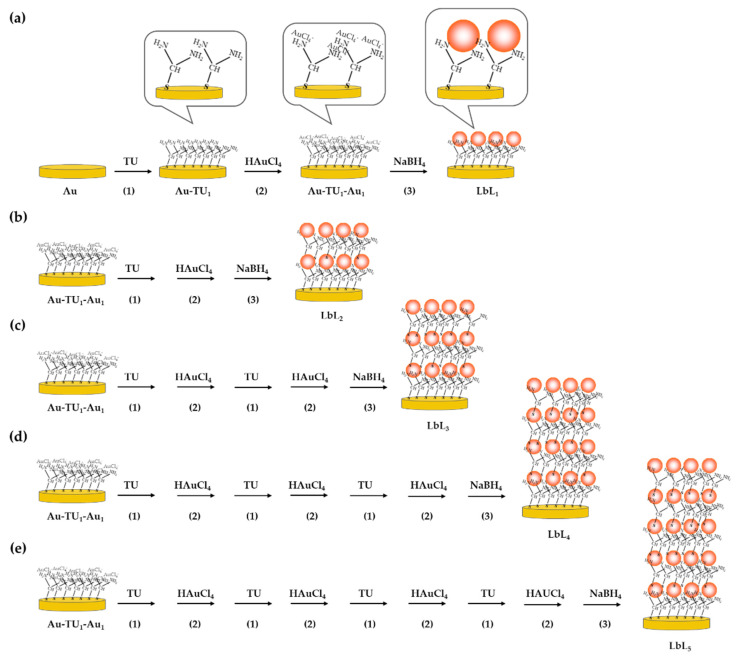
Preparation of layer-by-layer self-assembly of TU and HAuCl_4_ on the electrode surface (**a**) preparation of 1 layer and (**b**–**e**) preparation of 2, 3, 4, and 5 layers, respectively.

**Figure 2 sensors-22-00099-f002:**
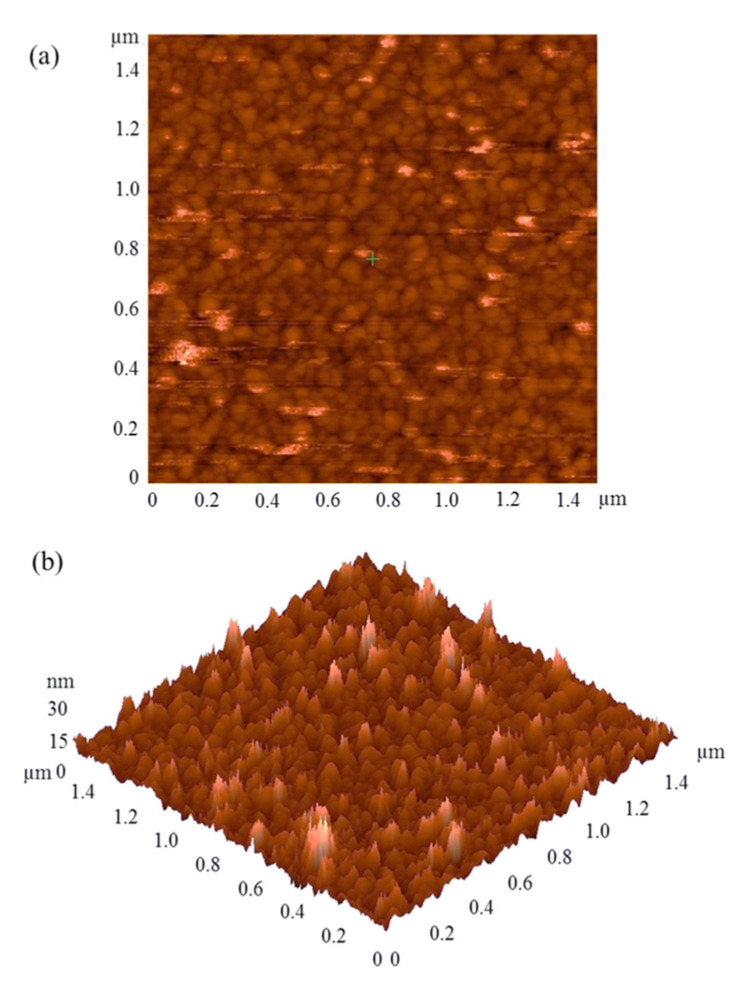
(**a**) Two- and (**b**) three-dimensional AFM images of the electrode fabricated by thermal evaporation technique.

**Figure 3 sensors-22-00099-f003:**
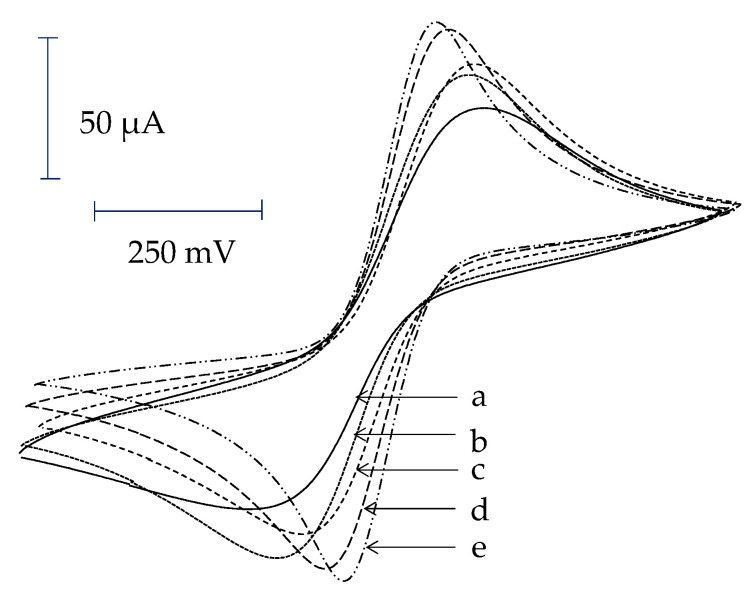
Cyclic voltammograms of electrodes modified with different number of layers (**a**) bare gold electrode, (**b**) LbL_1_, (**c**) LbL_2_, (**d**) LbL_3_, and (**e**) LbL_4_.

**Figure 4 sensors-22-00099-f004:**
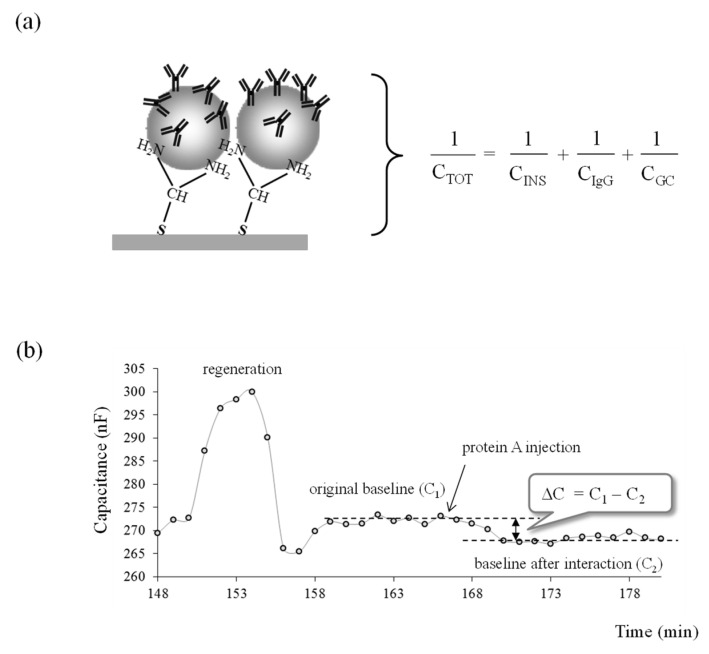
(**a**) Schematic representation of an assay cycle with a capacitive biosensor, where the total capacitance is described by several capacitors in series. A total capacitance (*C_TOT_*) has three components: (1) the capacitance of the insulating layer (C_INS_), (2) the capacitance of the human IgG layer (C_IgG_), and (3) the capacitance of the Gouy-Chapman diffuse layer (C_GC_); (**b**) capacitance registration from PA-human IgG binding interaction.

**Figure 5 sensors-22-00099-f005:**
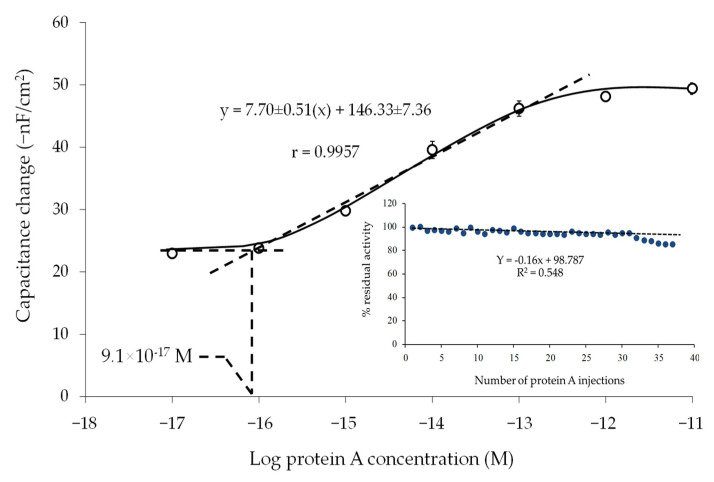
Capacitance change vs. logarithm of PA concentration; insert shows percentage residual activity of the electrode after regeneration.

**Table 1 sensors-22-00099-t001:** Sensitivity of electrodes modified with SAM of TU and layer-by-layer of TU and AuNPs.

Electrode Modifications	Sensitivity (−nF/cm^2^/log M)
SAM 2 h	2.31 ± 0.23
SAM 24 h	5.11 ± 0.23
LbL_1_	5.48 ± 0.31
LbL_2_	7.71 ± 0.30
LbL_3_	7.66 ± 0.43
LbL_4_	7.69 ± 0.37
LbL_5_	6.64 ± 0.42

**Table 2 sensors-22-00099-t002:** Comparison of the present study with related works focused on layer-by-layer immobilization techniques.

Analyte	Detection Method	Detection Range	LOD	References
carcinoma antigen 125	quartz crystal microbalance	1.5–180 U ml^−1^	0.5 mL^−1^	[33]
glucose	amperometric	0.5–16 mM	7.0 µM	[34]
DNA	impedance	1.0 × 10^−12^ to 1.0 × 10^−6^ M	3.1 × 10^−13^ M	[36]
virus	quartz crystal microbalance	2 × 10^0^ to 2 × 10^6^ PFU mL^−1^	2 PFU mL^−1^	[38]
protein A	capacitive	1 × 10^−16^ to 1 × 10^−13^ M	9 × 10^−17^ M	This work

## Data Availability

Not applicable.

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
