# Peer review of "A Sensitive Capacitive Biosensor for Protein a Detection Using Human IgG Immobilized on an Electrode Using Layer-by-Layer Applied Gold Nanoparticles"

_sensors, 2021, doi:10.3390/s22010099_

Round 1
Reviewer 1 Report
This manuscript reports a layer-by-layer (LbL) fabrication of gold nanoparticle layers on an electrode for capacitive immunosensing of protein A. While the sensors were proven to be functional, I have several concerns regarding the Ms which should be addressed before this Ms can eventually be accepted for publication.
- The fabrication method of AuNP layers on electrodes, as described in this Ms, for subsequent immunobinding and sensing, has been extensively reported in literature during the last decade or so. The key concept in this Ms, the layer-by-layer (deposition) seems incremental and only marginally improved sensitivity (Table 1). Therefore, the novelty and significance of this Ms is very ambiguous to me. The authors need to do a much better job to advocate those in the introduction and discussion sections.
 - There is a lack of characterization of the proposed multi-layer structure. How exactly did the surface morphology and roughness change when more layers were deposited? Additional characterization (e.g., more AFM, SEM imaging) could be helpful to provide more info and justify the difference in sensitivity and IgG immobilization yield.
 - The authors showed good regeneration experiments. But it may be of interest to also test the storage stability (in terms of loss of sensitivity) for extended periods, potentially in different conditions such as buffer/air/vacuum, to show the long-term stability of the LbL structures.
 - It's necessary to include more detailed, quantitative comparison between the detection range and LOD reported in this Ms and those already in the literature.

Reviewer 2 Report
In this work, the authors developed a novel electrochemical biosensor for protein a detection with IgG and layer-by-layer gold nanoparticles. The conducted characterizations were satisfactory and analytical performances were good. Therefore, I recommend acceptance of this work after minor revision.
- AFM images should be better presented with colors.
 - English language grammars should be checked. For example, “Chloroauric acid (HAuCl4) and thiourea (TU; NH2CSNH2) was purchased” on line 95, page 2.
 - The regeneration of 30 times should be supported by detailed data.
 - Selectivity should be further studied by introducing different kinds of proteins.

Round 2
Reviewer 1 Report
The authors answered most of my questions adequately. Generally, I recommend acceptance of this Ms by Sensors. But I still have reservations about their answer to my question 2. CV is no substitute for surface characterizations, especially when such surface characterizations are to be used to verify their observations with CV. I hope this can be fixed in future studies of the authors.